# Effect of Percutaneous Endoscopic Gastrostomy on Quality of Life after Chemoradiation for Locally Advanced Nasopharyngeal Carcinoma: A Cross-Sectional Study

Yun Xu [1,†], Hewei Peng [2,†], Qiaojuan Guo [1,3], Lanyan Guo [4], Xiane Peng [2,5,*] and Shaojun Lin [1,3,*]

1   Department of Radiation Oncology, Clinical Oncology School of Fujian Medical University, Fujian Cancer Hospital, Fuzhou 350000, China
2   Department of Epidemiology and Health Statistics, School of Public Health, Fujian Medical University, Fuzhou 350000, China
3   Fujian Key Laboratory of Translational Cancer Medicine, Fuzhou 350000, China
4   School of Medical Imaging, Fujian Medical University, Fuzhou 350000, China
5   Key Laboratory of Ministry of Education for Gastrointestinal Cancer, Fujian Medical University, Fuzhou 350000, China
*   Correspondence: fmuxe@163.com (X.P.); linshaojun@yeah.net (S.L.)
†   These authors contributed equally to this work.

**Abstract:** (1) Background: Prophylactic percutaneous endoscopic gastrostomy (PEG) maintained nutritional status and improved survival of patients with locally advanced nasopharyngeal carcinoma (LA-NPC). However, the role of PEG in patients' quality of life (QoL) is still controversial. We aimed to investigate the effect of PEG on the QoL of patients with LA-NPC without progression. (2) Methods: Patients with LA-NPC between 1 June 2010 and 30 June 2014 in Fujian Cancer Hospital were divided into PEG and non-PEG groups. The QoL Questionnaire core 30 (QLQ-C30), incidence of adverse effects, weight, and xerostomia recovery were compared between the two groups of patients without progression as of 30 June 2020. (3) Results: No statistically significant difference in the scores of each QLQ-C30 scale between the two groups ($p > 0.05$). The incidence of xerostomia was higher in the PEG group than in the non-PEG group ($p = 0.044$), but the association was not seen after adjusting for gender, age, T, and N stage (OR: 0.902, 95%CI: 0.485–1.680). No significant difference in the incidence of other adverse effects as well as in weight and dry mouth recovery ($p > 0.05$). (4) Conclusion: PEG seems not to have a detrimental effect on long-term Qol, including the self-reported swallowing function of NPC patients without progressive disease.

**Keywords:** nasopharyngeal carcinoma; percutaneous endoscopic gastrostomy; intensity-modulated radiotherapy; nutritional support; quality of life

## 1. Introduction

Nasopharyngeal carcinoma (NPC) is a malignant epithelial cancer prevalent in southern China. Radiotherapy is the primary treatment for NPC owing to its high sensitivity to radiation and the complex anatomy of the nasopharynx [1]. Most patients with NPC are in the locally advanced stage when diagnosed, at which point intensity-modulated radiotherapy (IMRT) with concurrent chemoradiotherapy (CCRT) is the standard treatment [2,3]. IMRT can increase the survival of patients and decrease damage to normal tissues [4–7]. The quality of life (QoL) and late toxicities have attracted more and more attention, along with improved survival.

As a result of the side effects of radiotherapy and chemotherapy, patients with locally advanced nasopharyngeal carcinoma (LA-NPC) commonly suffer from varying degrees of malnutrition and poor QoL during CCRT [8,9]. Several studies have demonstrated the link between poor nutritional status and a lower rate of survival among patients with NPC [10–12]. Moreover, nutritional status proved to be independently associated with

QoL in cancer patients [13,14]. As a result, enteral nutritional support is an effective tool for patients with LA-NPC to preserve their nutritional status during treatment, thereby ensuring a smooth treatment progression.

The nasogastric tube, percutaneous endoscopic gastrostomy (PEG), and surgical gastrostomy are commonly used methods to provide enteral nutritional support. Nasogastric tubes are only appropriate for patients who are unable to eat by mouth for a brief period of time (less than 30 days) but need nutritional support, while PEG is appropriate for patients who need long-term (more than 30 days) nutritional support [15,16]. PEG is less invasive, easier to handle, has fewer complications, and is less costly than surgical gastrostomy, making it more accessible to patients [17]. Our previous study found that prophylactic PEG prior to CCRT, as well as aggressive enteral nutritional support, maintained the nutritional status of patients with LA-NPC during CCRT and improved treatment completion rates [18]. These advantages can be translated into survival advantages for N3 NPC patients [19]. However, the role of PEG in patients' QoL is still controversial [20–29]. Prophylactic PEG before radiotherapy increases QoL in patients with head and neck cancer [21–24]. Some research, however, indicates that prophylactic gastrostomy placement prior to radiotherapy for patients with head and neck cancer is associated with a higher incidence of dysphagia and a greater reliance on PEG nutritional support [25–29]. Moreover, studies on the impact of PEG on QoL in NPC patients are lacking. Hence, this study aims to investigate the impact of PEG on patients' long-term QoL.

## 2. Materials and Methods

### 2.1. Patients and Study Design

Patients with pathologically confirmed progressive stage of primary nasopharyngeal carcinoma admitted to Fujian Cancer Hospital between 1 June 2010 and 30 June 2014 were included in this retrospective study. One hundred and thirty-three NPC patients who had voluntarily opted for prophylactic PEG feeding before receiving CCRT and 133 non-PEG patients who were matched based on age, gender, and tumor, node, and metastases level were recruited first [18]. Further exclusion criteria were as follows: (1) By 30 June 2020, patients who had died, had a recurrence or metastasis. (2) Patients under the age of 18 at the time of the initial consultation. (3) Patients who failed to complete the QOL questionnaires. As shown in Figure 1, a total of 148 NPC patients were finally enrolled in this study. The research was approved by the Fujian Cancer Hospital's Ethics Review Committee, and all participants signed a written informed consent form.

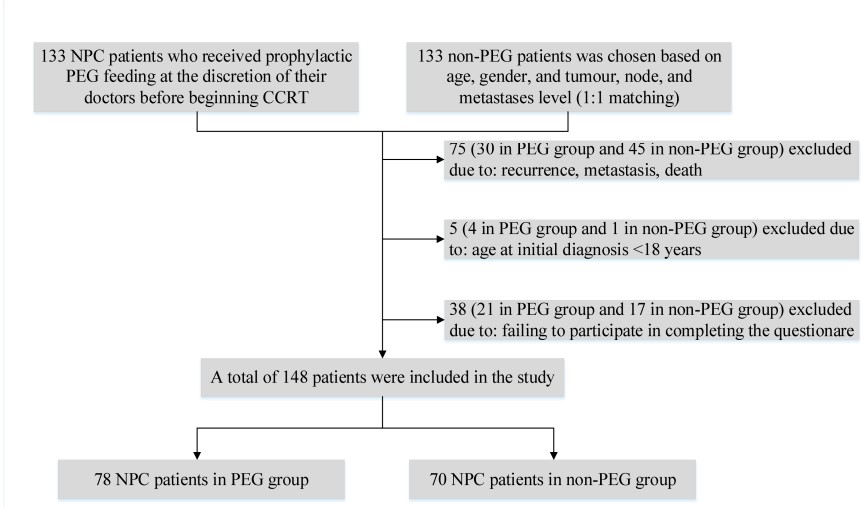

**Figure 1.** Flowchart of the study population. NPC, nasopharyngeal carcinoma; PEG, percutaneous endoscopic gastrostomy; CCRT, concurrent chemoradiotherapy.

### 2.2. Radiation Treatment and Chemotherapy

All patients received IMRT in combination with systemic chemotherapy. The radiotherapy was at a total dose of 68.8–81.75 Gy (median, 70 Gy) in 31–36 fractions (median, 33 fractions) for the primary tumor site. Except for three patients who received only two CCRT cycles, all patients were treated with cisplatin-based neoadjuvant chemotherapy (NACT) and CCRT. Detailed descriptions of IMRT and chemotherapy regimens have been published previously [18].

### 2.3. PEG Placement and Nutritional Support

All patients were free of severe stomach and other gastrointestinal lesions, without any past history of aggressive liver disease, hepatic or kidney dysfunction, congestive heart failure, chronic malignancy, dementia, respiratory failure, and coma. The pull method was used to place all PEG tubes [30]. PEG tubes were placed before the start of radiotherapy and were removed only after the acute mucositis had disappeared, and the patient was able to consume enough food orally. PEG tubes were removed approximately 4–6 weeks after the completion of radiotherapy in PEG patients. At the beginning of these conditions, patients were given enteral nutritional support. If the condition worsens, parenteral nutrition may be considered. The protocol was determined by a clinical dietitian to suit the individual needs of the patient and is adjusted as needed during treatment. More details about PEG and nutritional support have been shown in our previous studies [18,19]. All patients and their families were fully informed of the potential side effects of radiotherapy, the efficacy of PEG, and its expected advantages, as well as possible risks before treatment.

### 2.4. Data Collection

The late adverse effects of radiotherapy were evaluated in accordance with CTCAE 4.0. To assess the QoL of patients, we used the validated and internationally accepted European Organization for Research and Treatment of Cancer Quality of Life Questionnaire Version 3 (EORTC QLQ-C30) [31]. The questionnaire comprises 30 questions and is divided into 15 domains. There are five multi-item functional scales (physical, cognitive, emotional, social, and role functions), as well as three multi-item symptom scales (fatigue, pain, and nausea/emesis) and general QoL. Six single-item scales concerning dyspnea, insomnia, appetite loss, constipation, diarrhea, and financial difficulties are also included. The items were graded on a 1–4 scale, with the exception of the general QoL issue, which was scored on a 1–7 scale. The mean score for each scale was calculated and transformed into a value between 0 and 100. Higher scores for functioning and general QoL suggest better functioning and general QoL, whereas higher scores for symptoms indicate worse outcomes. To optimize the response rate, questionnaires were evaluated by telephone interviews conducted by the same professional training investigator. Assessment of late adverse effects and EORTC QLQ-C30 were collected between September 10 and November 10 in 2020. The time that passed between the CCRT and the questionnaire (median (P25, P75)) was 94.33 (85, 104.38) and 93.17 (82.45, 101.65) months in the non-PEG and PEG group, respectively.

### 2.5. Statistical Analyses

The baseline characteristics of subjects were analyzed using the *t* test for normal continuous variables and Nonparametric Kruskal–Wallis test for non-normal continuous variables. The Chi-Square test or Fisher exact probability method was used to analyze qualitative data. Multivariate ordinary logistic analysis was performed to evaluate the association between PEG and xerostomia. SPSS, version 19.0.0.1 (IBM SPSS, 2010, Chicago, IL, USA) was used for statistical analyses. All *p* values were two-tailed, and $p < 0.05$ was considered statistically significant.

## 3. Results

### 3.1. Baseline Characteristics of the Study Population

A total of 148 NPC patients (78 in the PEG group and 70 in the non-PEG group) were included in the study based on the inclusion criteria. The male patients accounted for 107 (72.3%) of the total. The entire cohort's mean age at initial diagnosis was $43.27 \pm 11.34$ years. In terms of gender, age, educational level, pathological type, clinical stage, T stage, N stage, and chemotherapy regimen, there were no statistically significant differences between the two groups (each $p > 0.05$). All details are provided in Table 1.

**Table 1.** Baseline characteristics of the study population.

| Variable | Non-PEG | PEG | *p* Value |
|---|---|---|---|
| Gender, *n* (%) | | | 0.714 |
| Male | 52 (74.3) | 55 (70.5) | |
| Female | 18 (25.7) | 23 (29.5) | |
| Age, mean $\pm$ SD | $42.3 \pm 12.1$ | $44.1 \pm 10.7$ | 0.310 |
| Educational level, *n* (%) | | | 0.193 |
| Primary school and less than | 18 (25.7) | 21 (26.9) | |
| Junior middle and high school | 37 (52.9) | 31 (39.7) | |
| Junior college or above | 15 (21.4) | 26 (33.3) | |
| Pathology subtype, *n* (%) | | | 0.599 |
| Keratinizing squamous | 2 (2.9) | 2 (2.6) | |
| Non-keratinizing undifferentiated squamous | 64 (91.4) | 68 (87.2) | |
| Non-keratinizing differentiated squamous | 4 (5.7) | 8 (10.3) | |
| Clinical stage, *n* (%) | | | 0.612 |
| III | 42 (60.0) | 41 (52.6) | |
| IVA | 18 (25.7) | 22 (28.2) | |
| IVB | 10 (14.3) | 15 (19.2) | |
| T stage, *n* (%) | | | 0.632 |
| T1 | 4 (5.7) | 8 (10.3) | |
| T2 | 13 (18.6) | 16 (20.5) | |
| T3 | 33 (47.1) | 30 (38.5) | |
| T4 | 20 (28.6) | 24 (30.8) | |
| N stage, *n* (%) | | | 0.431 |
| N0 | 1 (1.4) | 4 (5.1) | |
| N1 | 18 (25.7) | 15 (19.2) | |
| N2 | 41 (58.6) | 44 (56.4) | |
| N3 | 10 (14.3) | 15 (19.2) | |
| Regiments of CCRT | | | 0.137 |
| Single agent | 67 (95.7) | 69 (88.5) | |
| Two drugs | 3 (4.3) | 9 (11.5) | |

PEG, percutaneous endoscopic gastrostomy; CCRT, concurrent chemoradiotherapy.

### 3.2. Comparison of Late Toxicities between Non-PEG and PEG Groups

As shown in Table 2, the PEG group had a higher incidence of xerostomia than the non-PEG group (51.7% vs. 50%), and the difference was statistically significant ($p = 0.044$). Multivariate ordinary logistic analysis was further performed to investigate the relationship between PEG and xerostomia. After adjusting for potential confounding factors such as sex, age, T stage, and N stage, no statistically significant association was found between PEG and xerostomia (OR: 0.902, 95%CI: 0.485–1.680, $p = 0.783$) (Figure 2). There were no statistically meaningful variations between the two groups of patients in terms of other distant adverse effects (all $p > 0.05$). The most frequent late toxicity events were hearing loss (68.9%) and xerostomia (50.7%), as shown in Table 2.

**Table 2.** Comparison of late toxicities between non-PEG and PEG groups.

| Variables | All, *n* (%) | Non-PEG | | | | PEG | | | | *p* Value [a] |
|---|---|---|---|---|---|---|---|---|---|---|
| | | Grade 0, *n* (%) | Grade 1, *n* (%) | Grade 2, *n* (%) | Grade 3, *n* (%) | Grade 0, *n* (%) | Grade 1, *n* (%) | Grade 2, *n* (%) | Grade 3, *n* (%) | |
| Neck fibrosis, *n* (%) | 62 (41.9) | 47 (67.1) | 19 (27.1) | 4 (5.7) | 0 (0) | 39 (50.0) | 35 (44.9) | 2 (2.6) | 2 (2.6) | 0.052 |
| Xerostomia, *n* (%) | 75 (50.7) | 35 (50.0) | 22 (31.4) | 13 (18.6) | 0 (0) | 38 (48.7) | 32 (41.0) | 5 (6.4) | 3 (3.8) | 0.044 |
| Worst hearing, *n* (%) | 102 (68.9) | 19 (27.1) | 42 (60.0) | 4 (5.7) | 5 (7.1) | 27 (34.6) | 42 (53.8) | 5 (6.4) | 4 (5.1) | 0.757 |
| Tinnitus, *n* (%) | 63 (42.6) | 40 (57.1) | 21 (30.0) | 6 (8.6) | 3 (4.3) | 45 (57.7) | 22 (28.2) | 11 (14.1) | 0 (0) | 0.224 |
| Trismus, *n* (%) | 10 (6.8) | 64 (91.4) | 5 (7.1) | 0 (0) | 1 (1.4) | 74 (94.9) | 4 (5.1) | 0 (0) | 0 (0) | 0.495 |
| Dysphagia, *n* (%) | 46 (31.1) | 50 (71.4) | 14 (20.0) | 6 (8.6) | 0 (0) | 52 (66.7) | 22 (28.2) | 3 (3.8) | 1 (1.3) | 0.335 |
| Dysarthria, *n* (%) | 11 (7.4) | 65 (92.9) | 3 (4.3) | 0 (0) | 2 (2.9) | 72 (92.3) | 5 (6.4) | 1 (1.3) | 0 (0) | 0.329 |
| Chewing, *n* (%) | 22 (14.9) | 62 (88.6) | 1 (1.4) | 7 (10.0) | 0 (0) | 64 (82.1) | 2 (2.6) | 8 (10.3) | 4 (5.1) | 0.260 |
| Hoarseness, *n* (%) | 9 (6.1) | 68 (97.1) | 1 (1.4) | 0 (0) | 1 (1.4) | 71 (91.0) | 3 (3.8) | 1 (1.3) | 3 (3.8) | 0.451 |
| Tongue dysfunction, *n* (%) | 5 (3.3) | 67 (95.7) | 1 (1.4) | 1 (1.4) | 1 (1.4) | 76 (97.4) | 2 (2.6) | 0 (0) | 0 (0) | 0.480 |

[a] Fisher exact probability method. PEG, percutaneous endoscopic gastrostomy.

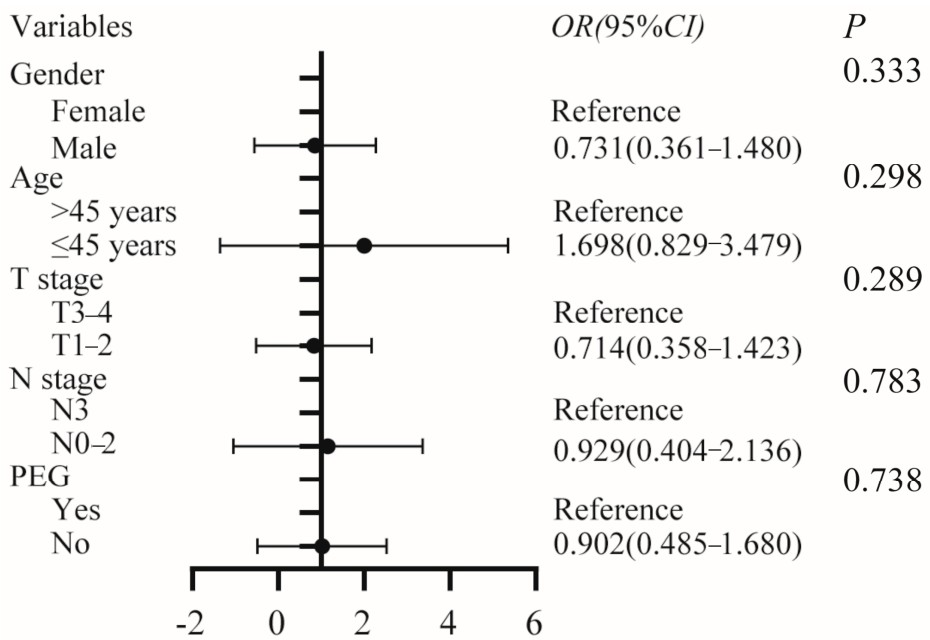

**Figure 2.** Multivariate ordinary logistic analysis of the association between PEG and xerostomia, adjusted for gender, age, T stage, and N stage. PEG, percutaneous endoscopic gastrostomy.

### 3.3. Results of the EORTC QLQ-C30

Data for the EORTC QLQ-C30 scales are presented in Table 3. There were no significant differences between PEG and non-PEG group in the score of EORTC QLQ-C30 scales.

**Table 3.** Comparison of quality of life between non-PEG and PEG groups.

| Variables | Non-PEG, Median (Range) | PEG, Median (Range) | All, Mean ± SD | *p* Value |
|---|---|---|---|---|
| Global health status/QoL | 83 (8–100) | 83 (25–100) | 83.8 ± 16.1 | 0.826 |
| Physical functioning | 100 (53–100) | 100 (53–100) | 94.2 ± 11.9 | 0.322 |
| Role functioning | 100 (0–100) | 100 (0–100) | 96.5 ± 15.5 | 0.633 |
| Emotional functioning | 100 (50–100) | 100 (25–100) | 89.5 ± 15.2 | 0.707 |
| Cognitive functioning | 83 (0–100) | 83 (33–100) | 83.0 ± 19.5 | 0.178 |

**Table 3.** *Cont.*

| Variables | Non-PEG, Median (Range) | PEG, Median (Range) | All, Mean $\pm$ SD | *p* Value |
|---|---|---|---|---|
| Social functioning | 100 (0–100) | 100 (0–100) | 90.1 $\pm$ 21.2 | 0.739 |
| Fatigue | 0 (0–33) | 0 (0–100) | 14.9 $\pm$ 21.7 | 0.169 |
| Nausea and vomiting | 0 (0–100) | 0 (0–50) | 0.9 $\pm$ 5.4 | 0.144 |
| Pain | 0 (0–100) | 0 (0–33) | 2.3 $\pm$ 10.4 | 0.249 |
| Dyspnea | 0 (0–67) | 0 (0–100) | 4.5 $\pm$ 15.4 | 0.859 |
| Insomnia | 0 (0–100) | 0 (0–100) | 14.0 $\pm$ 25.5 | 0.242 |
| Appetite loss | 0 (0–67) | 0 (0–100) | 4.73 $\pm$ 15.56 | 0.980 |
| Constipation | 0 (0–100) | 0 (0–67) | 5.86 $\pm$ 16.37 | 0.727 |
| Diarrhea | 0 (0–67) | 0 (0–0) | 3.83 $\pm$ 14.30 | 0.868 |
| Financial difficulties | 0 (0–67) | 0 (0–0) | 9.91 $\pm$ 21.46 | 0.683 |

PEG, percutaneous endoscopic gastrostomy; QoL, quality of life.

### 3.4. Comparison of Weight and Xerostomia Recovery between Non-PEG and PEG Groups

Compared with patients in the PEG group, more patients in the non-PEG group took more than a year to return to baseline weight and recover from xerostomia, but no statistically significant differences between groups were seen (all *p* > 0.05) (Table 4).

**Table 4.** Comparison of weight and xerostomia recovery between non-PEG and PEG groups.

| Variables | Non-PEG | PEG | *p* Value |
|---|---|---|---|
| Time to return to baseline weight | | | 0.425 |
| $\leq$1 years | 34 (48.6) | 43 (55.1) | |
| >1 years | 36 (51.4) | 35 (44.9) | |
| Time of recovery from xerostomia | | | 0.628 |
| $\leq$1 years | 19 (27.1) | 24 (30.8) | |
| >1 years | 51 (72.9) | 54 (69.2) | |

PEG, percutaneous endoscopic gastrostomy.

## 4. Discussion

Patients who underwent prophylactic PEG experienced significant improvements in nutritional status and QoL while also showing increased treatment adherence during radiotherapy. Nonetheless, among patients with head and neck cancer, the role of PEG in terms of long-term QoL and adverse effects is debatable. In this study, xerostomia was more common in the PEG group than in the non-PEG group, but the association was not seen after adjusted for gender, age, T, and N stage. The frequency of other adverse effects, such as dysphagia, did not vary statistically significantly between the two groups. There was no statistically significant difference between the two groups in terms of QoL. However, the patients' high general QoL scores showed that both groups of patients had a decent general QoL. To our knowledge, this is the first research that examines the impact of prophylactic PEG on long-term QoL and adverse effects in NPC patients. PEG does not appear to have a detrimental effect on long-term QoL, including swallowing function, according to our findings.

Similar results were observed in several head and neck cancer studies [22–24]. Axelsson et al. [22] used an EORTC QLQ-head and neck 35 scale and a five-level oral intake scale to test swallowing outcomes in a randomized study that included patients with head and neck cancer who were randomly assigned to one of two groups: Prophylactic PEG or nutritional support according to clinical practice. The patients' capacity to swallow foods did not vary between the groups, according to the findings. Prestwich et al. [23] retrospectively included 56 patients with head and neck cancer in two matched groups who received either a prophylactic gastrostomy tube (GT) or a nasogastric tube as required and used the MD Anderson Dysphagia Inventory questionnaire to assess swallowing outcomes. In line with our findings, there was no significant difference in long-term swallowing

function between the groups. Another study conducted by Prestwich et al. [24] showed the same results, as well.

However, some studies indicated that prophylactic PEG increases the risk of long-term dysphagia [25–28]. Patients who received prophylactic GT before treatment had a higher incidence of GT dependence and stricture diagnosis than those who did not. The authors hypothesized that the high incidence of long-term GT dependency in patients may be due to atrophy of the muscles that control the swallowing process [25]. Oozeer et al. [26] performed another analysis that yielded the same findings. Prophylactic PEG tubes were independent predictors of PEG tube dependency at least one year after treatment in patients with head and neck cancer who received definitive chemoradiation, according to a retrospective review [27]. A retrospective study [28] supports the hypothesis that patients treated with PEG feeding have higher severe late dysphagia than patients treated with reactive nasogastric feeding. The convenience of PEG placement, according to the authors, can deter patients from working hard to become nutritionally independent after therapy is completed. The opposite was found in our research. There was no significant difference between PEG and non-PEG groups in terms of long-term QoL, including dysphagia. Unlike the studies above, only NPC patients were included in our study. During radiotherapy, we encouraged patients in PEG groups to do swallowing exercises like drinking. In addition, the PEG tube was removed after the acute mucositis had resolved, allowing for adequate food intake orally (approximately four to six weeks after the end of radical radiotherapy). Moreover, to avoid interference with recurrence and metastasis, only patients without progression were included in our analysis.

Using the EORTC QLQ-C30 scale to assess the QoL of NPC patients who survived more than two years, a study included 216 NPC survivors found that these patients had a slightly high incidence of dry mouth, fatigue, hearing loss, depression, and anxiety, but had a good QoL [32]. Another randomized controlled trial [33] showed that the observation group (nutritional support) had a lower incidence of adverse effects and had better short-term outcomes and QoL than the control group, which was likely due to the patients' improved nutritional status. Of the 148 patients in our study, 102 (68.9%) had hearing loss, and 75 (50.7%) were troubled by xerostomia. Patients, however, had higher mean scores for overall QoL as well as the five major functions of somatic, social, task, emotional, and cognitive functioning, and lower scores for the remaining symptoms. The fact that all of our study participants received intensity-modulated radiotherapy may have contributed to their high QoL. Intensity-modulated radiotherapy, as compared to traditional radiotherapy, helps protect normal tissues, reduce the occurrence of long-term side effects, and increase patients' long-term QoL [34–37]. The other possible reason may be that the final analysis included only patients without progression.

There are several limitations to the current study. First, there was selection bias in this study since it was not a prospective randomized controlled trial, and the decision to conduct PEG was based on the patients' wishes. Second, investigators gathered information on patients' QoL mostly through telephone follow-up inquiries, resulting in information bias. Bias may be minimized to some extent in this study because the questionnaire was completed item by item by the same professionally qualified investigator during the telephone follow-up of the patients. Third, the limited sample size resulting from a single center and restrictive inclusion conditions may impose some limitations on generalization. Prospective multicenter studies with a large sample size are required to validate our findings in the future.

## 5. Conclusions

During concurrent chemoradiotherapy, prophylactic PEG enhanced the nutritional status of patients with LA-NPC, without any adverse consequences for long-term QOL, including self-reported swallowing function. It remains an active and effective nutritional intervention for patients with LA-NPC who are deficient in nutrition and are un-

able to successfully complete the treatment. Prospective studies are also required for further evidence.

**Author Contributions:** Conceptualization, S.L. and X.P.; Methodology, Y.X. and H.P.; Software, H.P. and X.P.; Validation, Y.X., Q.G., L.G. and S.L.; Formal Analysis, Y.X. and H.P.; Investigation, Y.X., H.P. and L.G.; Resources, Y.X., Q.G., L.G. and S.L.; Data Curation, Y.X., H.P., S.L. and X.P.; Writing—Original Draft Preparation, Y.X. and H.P.; Writing—Review & Editing, S.L. and X.P.; Visualization, S.L. and X.P.; Supervision, S.L. and X.P.; Project Administration, S.L.; Funding Acquisition, Y.X. and Q.G. All authors have read and agreed to the published version of the manuscript.

**Funding:** This work was funded by the Natural Science Foundation of Fujian Province, grant number 2019Y0061, the Training Program for Young and Middle-aged Backbone Talents of Fujian Provincial Health Care Commission, grant number 2020GGB013 and the Training Program for Young and Middle-aged Backbone Talents of Fujian Provincial Health Care Commission, grant number 2020GGA014, National Clinical Key Specialty Construction Program, and Fujian Provincial Clinical Research Center for Cancer Radiotherapy and Immunotherapy, grant number 2020Y2012.

**Institutional Review Board Statement:** The research was carried out in compliance with the Declaration of Helsinki and approved by the Fujian Cancer Hospital's Ethics Review Committee.

**Informed Consent Statement:** Written informed consent has been obtained from the patient(s) to publish this paper.

**Data Availability Statement:** The datasets used can be available from the corresponding author on reasonable request.

**Acknowledgments:** The authors would like to express their gratitude to all participants for their cooperation and to all staffs for recruiting subjects and their technical assistance.

**Conflicts of Interest:** The authors declare no conflict of interest.

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
