# Peer review of "Effect of Percutaneous Endoscopic Gastrostomy on Quality of Life after Chemoradiation for Locally Advanced Nasopharyngeal Carcinoma: A Cross-Sectional Study"

_curroncol, doi:10.3390/curroncol30010076_

Round 1
Reviewer 1 Report
The main point of the article is to show that PEG is improving the Quality of Life of the patient undergoing CCRT. The assumption that a poor QoL occurs during CCRT should be supported by more than two papers.
In the second part (Materials and Methods) a few more words should have been spent on the reason why PEG was placed in some patients rather than others.
In the same part of the paper, the amount of time PEG was maintained is not mentioned. There should be at least the minimum time, the maximum time and also the mean value of the time spent by the patients with the PEG.
The authors should be more specific about the time that passed between the CCRT and the questionnaire.
In the Results part the explanation of the multivariate analysis can be improved, in order to clear any doubt about xerostomia being more prevalent in PEG-patients.
For the rest, the results were presented in a clear and ordered way.
In the conclusions this paper is compared with other works about the dysphagia item. But the cited articles talk about head and neck cancer in a general way, not specific as the NPC. If possible, the comparison with other papers about this feature should be improved with works concerning the same site cancer.
In the end, better results might have been achieved if the sampling were bigger than 148 patients. In the conclusions, specifically in the part concerning the limitation of the study, a statement should be added about the possibility to have bigger cohorts.
Reviewer 2 Report
Thankyou for the opportunity to review this manuscript. I have the following comments:
1. abstract - conclusion statement needs to specify the finding is in NPC patients without progressive disease; and it should state "self reported swallowing function" - as this was not objectively measured in this study.
2. Introduction is well written with justified background for the research question. Please just add in a reference to support the statements recommending NG vs PEG duration <30 or >30days (eg ESPEN guidelines)
3. Methods - more details are required here as described below
i) Line 73 - if starting a sentence with a number - need to spell out
ii) Figure 1 - pls show the exclusions applied to each arm of the PEG group and non PEG group
iii) In line 74, you report the PEG was placed at discretion of the doctor, but in the discussion lines 228-229 it is stated it was placed according to patient wishes. Please clarify this.
iv) There is no description of other healthcare professionals who may have been involved in care - did patients have access to nursing staff, dietitian, speech pathology, and other allied health services. This could all impact on how symptoms/concerns were addressed/managed and therefore impact on QOL outcomes
v) Please advise on PEG complication rates (if possible) - as again the experience of PEG complications could impact on QOL outcomes. If data not available this needs to be listed as limitation in discussion.
vi) section 2.4 - there is no description of when data was collected. please outline timepoints data was collected at and duration of follow up
vii) line 106 - change "bodily" to "physical"
viii) the QOL tools used is not clear. The EORTC QLG-C30 - has 30 questions as you outline, which are all measured on scale 1-4, except for the QOL questions which are 1-7. However you then talk about there being 3 multi-item symptom scale for fatigue, pain and nausea/emesis which confused me (is this still part of the 30 questions or something else?). In addition, you then describe six single item scales being included. Were these taken from the QLQ-H&N35? or the updated QLQ-H&N43? And if so, why why only these items selected and not the full tool? or are they from a different QOL tool?
ix) I am also concerned how the questionnaires were completed and if there is any bias here - the methods state in lines 114-115 "questionnaires were evaluated by telephone interviews by the same professional training investigator" which sounded fine. But then in the discussion lines 231-233 it states "questionnaire was filled out by same professionally qualified investigator after interviewing the patients" - so doe that mean the questions were not directly asked by the interviewer and responded to by the participants? and the forms were completed afterwards based on an "unstructured interview" with the patient. Please clarify exactly how these questionnaires were administered.
x) There is no mention in the methods as to what tools were used to assess late toxicity or adverse events that were reported on in table 2. or what timepoints these were assessed at? Please add this detail and information to the methods.
4. Results - the manuscript template is still showing up in line 125-127
5. Tables - for ease of reading, please provide data to 1dp (eg age in table 1, and the mean/sd in table 3) (keep p values to 3dp). Please also report in this format in the text - see line 131 where age is reported.
6. Table 1 - Suggest bold type for the variable headings to help readability. Add capital letters to the variable descriptions for Education Level. ie Primary and Junior
7. Figure 2 - add p value, and also insert to text in lines 142
8. Results - line 145-146 just reports the data in the table. Suggest trying to highlight key findings here eg The most frequent late toxicity events were hearing loss (68.9%) and xerostomia (50.7%) as shown in Table 2. Section 3 should perhaps start with this, and then go on to describe the differences between PEG and nonPEG groups after that.
9. Table 2 - there is a footnote "a" for Fisher Exact method - but could not see this footnote used in the Table. Has it been omitted in error or is it not needed?
10. Results lines 153-156 - again this simply reports on the exact data in the table. I suggest deleting, as the reader can just get this from the table.
11. table 3 - just adjust to showing the mean/sd data to 1dp
12. Discussion - overall well written to discuss study findings n context of the literature. Couple of minor comments:
i) line 201 - new abbreviation used here - R-NG. Please spell out.
ii) line 207-209 - re PEG removal criteria - was any consideration given to patients nutritional status? this should also be discussed further.
13. Conclusion - as per comment on the abstract -just need to specify the dysphagia findings were based on patient reported swallowing function (as not measured objectively in this study)
